# Self-Distillation Enables Continual Learning

**Idan Shenfeld** [1 2]  **Mehul Damani** [1]  **Jonas Hübotter** [3]  **Pulkit Agrawal** [1 2]

## Abstract

Continual learning, enabling models to acquire new skills and knowledge without degrading existing capabilities, remains a fundamental challenge for foundation models. While on-policy reinforcement learning can reduce forgetting, it requires explicit reward functions that are often unavailable. Learning from expert demonstrations, the primary alternative, is dominated by supervised fine-tuning (SFT), which is inherently off-policy. We introduce **Self-Distillation Fine-Tuning (SDFT)**, a simple method that enables on-policy learning directly from demonstrations. SDFT leverages in-context learning by using a demonstration-conditioned model as its own teacher, generating on-policy training signals that preserve prior capabilities while acquiring new skills. Across skill learning and knowledge acquisition tasks, SDFT consistently outperforms SFT, achieving higher new-task accuracy while substantially reducing catastrophic forgetting. In sequential learning experiments, SDFT enables a single model to accumulate multiple skills over time without performance regression, establishing on-policy distillation as a practical path to continual learning from demonstrations. Code and Datasets are available at http://idanshenfeld.com/SDFT.

## 1. Introduction

Foundation models have achieved remarkable success in recent years, powering AI applications across language, vision, robotics, and beyond. However, despite their impressive capabilities, today's AI systems remain static after deployment. While they can adapt their behavior at inference time through mechanisms such as retrieval or prompting, they do not update their parameters to acquire new skills, internalize new knowledge, or improve from experience.

[1]MIT [2]Improbable AI Lab [3]ETH Zurich. Correspondence to: Idan Shenfeld <idanshen@mit.edu>.

*Proceedings of the 43rd International Conference on Machine Learning*, Seoul, South Korea. PMLR 306, 2026. Copyright 2026 by the author(s).

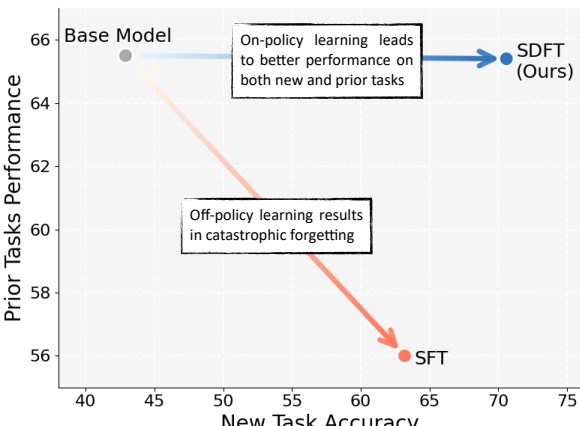

*Figure 1.* Supervised Fine-Tuning (SFT) is commonly used to learn from expert demonstration datasets, but its off-policy nature leads to catastrophic forgetting of general capabilities. We introduce Self-Distillation Fine-Tuning (SDFT), which turns expert demonstrations into on-policy learning signals by using a demonstration-conditioned version of the model as its own teacher. In this way, SDFT enables true continual learning with the model improving on new tasks as they arise without regressing existing capabilities.

To enable the next generation of foundation models, we must solve the problem of continual learning: enabling AI systems to keep learning and improving over time, similar to how humans accumulate knowledge and refine skills throughout their lives (Hassabis et al., 2017; De Lange et al., 2021).

A growing body of recent work has highlighted the importance of on-policy learning for continual learning. When models learn from data generated by their current policy, they exhibit substantially reduced catastrophic forgetting compared to off-policy alternatives (Shenfeld et al., 2025; Chen et al., 2025). To date, most successful on-policy approaches have been developed in the context of reinforcement learning (RL), where feedback is provided through an explicit reward function. However, in many real-world settings such rewards are unavailable or difficult to specify. Instead, learning typically proceeds from datasets of expert demonstrations. The dominant paradigm in this regime is supervised fine-tuning (SFT), which trains the model to imitate expert actions under a fixed, offline data distribution. While simple and scalable, SFT is inherently off-policy, and prior work has shown that sequential SFT can lead to poor

generalization and severe catastrophic forgetting when models are adapted to new tasks or domains (Kirkpatrick et al., 2017; Li & Hoiem, 2017). This tension raises a fundamental challenge for continual learning: *how can we obtain the benefits of on-policy learning when only demonstrations are available?*

The challenges of off-policy learning can, in principle, be overcome by first learning a reward function from demonstrations (i.e., Inverse Reinforcement Learning or IRL), and then performing on-policy RL (Ng et al., 2000; Abbeel & Ng, 2004). While IRL is conceptually elegant, effectively recovering rewards typically requires strong priors over the reward structure, which has limited its practical adoption to settings where such assumptions are justified, such as RLHF (Peng et al., 2018; Stiennon et al., 2020).

Rather than inferring an explicit reward function, we propose Self-Distillation Fine-Tuning (SDFT), an on-policy distillation (Ross et al., 2011; Agarwal et al., 2024) framework for learning directly from demonstrations. SDFT relies on the observation that large pretrained models exhibit strong in-context learning—the ability to adapt their behavior when conditioned on examples, without parameter updates (Brown et al., 2020). We exploit this property by using the same model in two roles: a teacher, conditioned on both the task input and an expert demonstration, and a student, conditioned only on the task input. Training distills the teacher's predictions into the student on trajectories generated by the student itself, yielding on-policy updates that incorporate information from demonstrations without explicit reward inference or offline imitation.

We evaluate SDFT in two continual learning settings: *skill learning*, where demonstrations are used to improve performance on a task, and *knowledge acquisition*, where new information must be incorporated into the model. Across both settings, SDFT provides stable on-policy updates that enable learning while substantially reducing catastrophic forgetting compared to supervised learning. Consistent with prior work on on-policy learning (Ross et al., 2011; Chu et al., 2025), SDFT also improves generalization both in-distribution and out-of-distribution, making it beneficial even in settings where retaining prior capabilities is not the primary objective. In a sequential learning experiment involving three distinct skills, SDFT enables a single model to acquire each skill in turn while preserving performance on previously learned skills as well as on unrelated, pre-existing capabilities — demonstrating that continual learning from demonstrations is possible.

## 2. Related Work

**Off-policy versus On-policy Learning.** A long line of work highlights the advantages of on-policy learning, i.e.,

training on trajectories induced by the model itself, over off-policy learning. The seminal result of Ross et al. (2011) shows that off-policy imitation learning suffers from compounding errors at inference time, as the learned policy drifts away from the states covered in the demonstrations, errors accumulate rapidly, a failure mode that on-policy algorithms avoid by continually training under their own state distribution. More recent empirical studies reinforce this distinction. Models fine-tuned with on-policy RL have been shown to generalize better beyond the training distribution (Agarwal et al., 2024; Han et al., 2025; Chu et al., 2025; Li et al., 2025) and transfer more effectively to related tasks (Huan et al., 2025) than models trained purely off-policy. In continual learning settings, on-policy updates also reduce catastrophic forgetting when adapting to new tasks (Shenfeld et al., 2025; Lai et al., 2025). These findings collectively motivate our goal - to enable on-policy learning from demonstrations, thereby retaining the benefits of on-policy RL while avoiding the need for explicit reward engineering.

**Inverse Reinforcement Learning.** IRL (Ng et al., 2000) provides a classical solution to the problem faced in many RL settings: the agent must learn a policy when no explicit reward function is available, only demonstrations. Rather than cloning the expert's actions, IRL seeks to infer the underlying reward for which those demonstrations would be optimal. This perspective avoids the issues of off-policy imitation learning, since the inferred reward can support on-policy updates (Xu et al., 2020). While this idea has deep theoretical appeal, traditional IRL methods are known not to scale well (Lazzati et al., 2024; Arora & Doshi, 2021).

A common thread across all successful IRL formulations is that they rely on strong structural assumptions to make the reward identifiable. Maximum-entropy IRL assumes that experts follow a soft-optimal Boltzmann policy (Ziebart et al., 2008; Wulfmeier et al., 2015); adversarial IRL methods (Ho & Ermon, 2016) assume that expert and learner trajectories can be distinguished by a classifier; and preference-based IRL methods, such as RLHF (Ziegler et al., 2019; Ouyang et al., 2022), assume access to pairs of positive–negative demonstrations. These priors are essential—without them, IRL is either ill-posed or too expensive to be practical. In our approach, rather than imposing an explicit learning reward function, we leverage the model's in-context learning to extract an on-policy learning signal.

**Context Distillation.** Our method also relates to the growing line of work on context distillation, in which a model conditioned on additional information acts as a teacher for a version of itself without that information (Bai et al., 2022; Snell et al., 2022). Prior approaches typically rely on offline distillation from static contexts, such as few-shot exam-

Self-Distillation Fine-Tuning

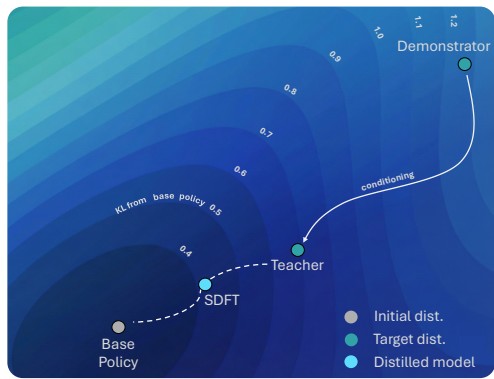

*Figure 2.* **(Left)** SDFT leverages a model's in-context learning ability to generate on-policy training signals. For each query $x$, the model acts in two roles. A student that is conditioned only on the query $P = \pi(\cdot|x)$ and the teacher, which is the same model conditioned on an expert demonstration $c$, producing a demonstration-aware distribution $Q = \pi(\cdot|x, c)$. Training minimizes the reverse KL divergence between the student and teacher, yielding on-policy updates. **(Right)** Conditioning the model on the expert demonstrations creates a teacher with an output distribution that is substantially closer to the base model, while maintaining the same new-task accuracy.

ples or behavioral guidelines, and supervise the student on trajectories drawn from the teacher's distribution. Our algorithm differs in two important ways. First, the distillation is on-policy: the student is trained under its own induced trajectory distribution, allowing the teacher to correct errors as they arise (Ross et al., 2011; Agarwal et al., 2024). Second, the context provided to the teacher is not a fixed prompt prefix but a *specific demonstration chosen for each query*. This dynamic, instance-wise conditioning enables the teacher to express fine-grained task intent, rather than a single global behavioral prior. Together, these differences allow context distillation to function not merely as a form of prompt compression but as an IRL-like mechanism that extracts and transfers the underlying reasoning induced by demonstrations.

## 3. Self-Distillation Fine-Tuning

Our approach builds on the framework of student-teacher distillation, where a student model is trained to match the behavior of a teacher model by minimizing the divergence between their output distributions. Traditionally, distillation uses separate models, typically a larger teacher and a smaller student (Hinton et al., 2015). Our key innovation is using *same* model as both teacher and student by exploiting its in-context learning abilities. Specifically, given a foundation model with policy $\pi$, the student is simply the base model without this conditioning $\pi_\theta(\cdot|x)$. The teacher is constructed by conditioning it on expert demonstrations $\pi(\cdot|x, c)$, where $x$ is the task prompt and $c$ is a demonstration, using the following simple prompt:

```
<Question>
This is an example for a response
to the question:
<Demonstration>
Now answer with a response of your
own, including the thinking
process:
```

We find that this prompt is sufficient to prevent the policy from outputting $c$ verbatim and instead elicits a response that reflects the model's understanding of the intent behind the demonstration, leveraging its in-context learning capabilities. See subsection 3.2 for further analysis of the conditioned policy's outputs.

As mentioned before, we hypothesize that on-policy learning is necessary for continual learning; therefore, we train the student using on-policy distillation from the teacher. For every prompt $x$, our algorithm, SDFT, samples responses from the student policy $y \sim \pi_\theta(\cdot|x)$ and minimizes the reverse Kullback-Leibler (KL) divergence between the student and the teacher distributions:

$$\mathcal{L}(\theta) = \mathbb{E}_{y \sim \pi_\theta(\cdot|x)} \left[ \log \frac{\pi_\theta(y|x)}{\pi(y|x, c)} \right] \quad (1)$$

Leveraging the autoregressive nature of the model, we decompose this objective into a token-level loss (see derivation in Tang & Munos (2025)) and take the gradient with respect to the student parameters $\theta$ while treating the teacher as fixed. This results in the following gradient estimator:

$$\nabla_\theta \mathcal{L}(\theta) =$$

$$\mathbb{E}_{y \sim \pi_\theta} \left[ \sum_t \sum_{y_t \in \mathcal{V}} \log \frac{\pi_\theta(y_t|y_{<t}, x)}{\pi(y_t|y_{<t}, x, c)} \nabla_\theta \log \pi_\theta(y_t|y_{<t}, x) \right]$$

where $\mathcal{V}$ is the token vocabulary. A critical component of SDFT is the parameterization of the teacher model used to

compute the likelihood ratios. Appendix A.3 includes an ablation regarding this design choice. Unless mentioned otherwise, we use an exponential moving average (EMA) of the student parameters for the teacher. A full detailed description of our algorithm can be found in Algorithm 1 in the appendix.

### 3.1. Self-Distillation as Inverse RL

Although we present our algorithm from a student-teacher distillation perspective, it can also be interpreted in the IRL framework, where it maximizes an implicit reward function. In the following section, we formally show that our self-distillation objective is mathematically equivalent to maximizing an implicit reward defined by the expert demonstrations and the model's in-context learning capabilities.

We begin with the standard formulation of trust-region-regularized reinforcement learning (Schulman et al., 2015), where the policy update in step $k + 1$ is constrained to stay close to the current policy $\pi_k$:

$$\pi_{k+1} = \max_{\pi} \mathbb{E}_{y\sim\pi}[r(y,x)] - \beta D_{\mathrm{KL}}(\pi(\cdot|x)\|\pi_k(\cdot|x))$$

For this objective, the optimal policy $\pi_{k+1}^*$ takes the known closed-form expression of a tilted distribution (Korbak et al., 2022; Rafailov et al., 2023):

$$\pi_{k+1}^*(y|x) \propto \pi_k(y|x)\,\exp(\frac{1}{\beta}\,r(y,x))$$

Rearranging this equation allows us to express the underlying reward as a function of the divergence between the optimal and previous policies:

$$r(y,x) = \beta\left[\log \pi_{k+1}^*(y|x) - \log \pi_k(y|x)\right] + C$$

In a standard IRL setting, $\pi_{k+1}^*$ is unknown. However, our key idea is that the model's own in-context learning capabilities provide a robust approximation of this optimal policy. We introduce our *In-Context Assumption* - given a demonstration $c$, the model conditioned on $c$ approximates the optimal next policy.

$$\pi_{k+1}^*(y|x) \approx \pi(y|x,c) \tag{2}$$

This substitution posits that the behavioral shift induced by observing a demonstration results in an optimal policy. Substituting it back:

$$r(y,x,c) = \log \pi(y|x,c) - \log \pi_k(y|x)$$

We drop $\beta$ and $C$ since linear transformations of reward do not affect the optimal policy (Sutton et al., 1998). While this defines a trajectory-level reward, our model has an autoregressive structure. Therefore, we decompose the reward into token-level rewards $r_t$ via token-level probabilities:

$$r_t(y_t\,|\,y_{<t},x,c) = \log \frac{\pi(y_t\,|\,y_{<t},x,c)}{\pi_k(y_t\,|\,y_{<t},x)} \tag{3}$$

and indeed for all $y$, we have $\sum_t r_t(y_t\,|\,y_{<t},x,c) = r(y,x,c)$. Finally, we demonstrate that optimizing the policy with respect to this reward is equivalent to the reverse-KL distillation used in our method. The policy gradient under the current policy $\pi_k$ is:

$$\nabla_\theta J(\pi_k) = \mathbb{E}_{y\sim\pi_k}\left[r(y,x,c)\nabla_\theta \log \pi_k(y|x)\right]$$

Substituting our derived reward from Equation 3:

$$\nabla_\theta J(\pi_k)$$
$$= \mathbb{E}_{y\sim\pi_k}\left[\sum_t \log \frac{\pi(y_t\,|\,y_{<t},x,c)}{\pi_k(y_t\,|\,y_{<t},x)}\nabla_\theta \log \pi_k(y_t\,|\,y_{<t},x)\right]$$

We observe that this is equivalent in expectation to the gradient of the reverse KL divergence $D_{\mathrm{KL}}(\pi_k(\cdot|x)\|\pi(\cdot|x,c))$ in Equation 3. Thus, our method can be viewed as an on-policy RL algorithm that maximizes rewards inferred by comparing the student's current behavior to its own "wiser," demonstration-aware counterpart.

### 3.2. Validating the ICL Assumption

The core hypothesis of SDFT can be seen as the assumption in Equation 2, which states that a model conditioned on an expert demonstration behaves like the (unknown) optimal policy for that task $\pi_{k+1}^*(y|x) \approx \pi(y|x,c)$ and therefore it can be a good teacher. The quality of this approximation depends on 2 conditions:

1. *Optimality*: The teacher's expected reward must match that of the unknown optimal policy:

$$\mathbb{E}_{y\sim\pi(y|x,c)}[r(y,x)] \approx \mathbb{E}_{y\sim\pi_{k+1}^*}[r(y,x)]$$

   In other words, samples drawn from the demonstration-conditioned policy should achieve near-maximal reward on the task.

2. *Minimal Deviation*: Due to the trust-region regularization in Equation 3.1, the optimal policy $\pi_{k+1}^*(y|x)$ is the one closest, in the KL sense, to the current model $\pi_k$ among all the ones that maximize reward. Thus, we require:

$$D_{\mathrm{KL}}\left(\pi(\cdot|x,c)\|\pi_k(\cdot|x)\right) \approx D_{\mathrm{KL}}\left(\pi_{k+1}^*(\cdot|x)\|\pi_k(\cdot|x)\right)$$

The second requirement, remaining close to the current policy, is crucial for practical viability. If the demonstration-conditioned teacher simply mimicked the example verbatim, it would deviate substantially from the base model, losing the benefits of on-policy learning. What makes the teacher valuable is that it produces new, task-appropriate behavior while remaining anchored to the base model.

**Empirical Validation.** While we cannot verify these conditions theoretically, we evaluate each empirically. We use

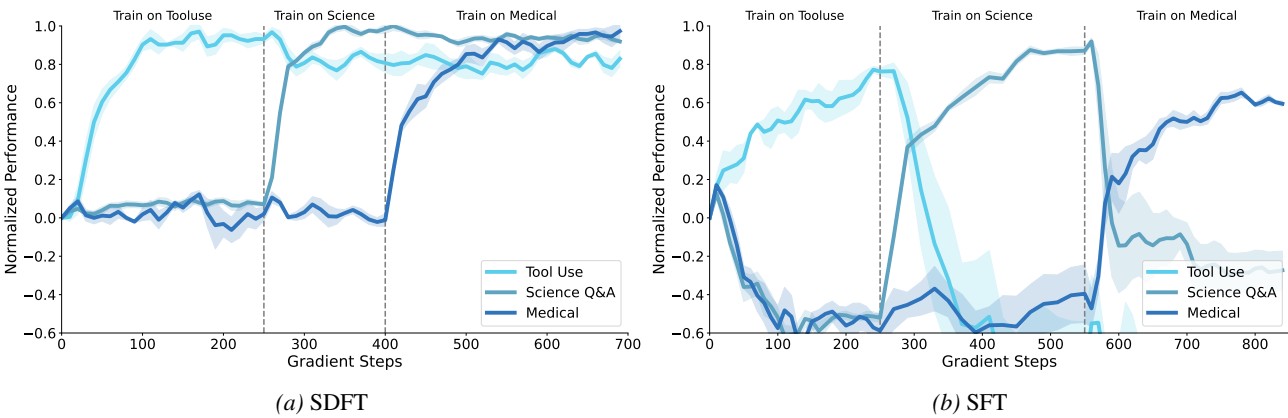

*Figure 3.* In a challenging continual learning experiment, where one model is trained sequentially on three different tasks, SDFT is able to learn each one while retaining performance on the others. In contrast, SFT performance on each task drops once it starts learning the next one. Performance is linearly normalized such that 0 corresponds to the base model accuracy on each one of the tasks, and 1 to the maximum accuracy obtained across both algorithms.

the Qwen-2.5-7B-Instruct model (Hui et al., 2024) as the base policy and ToolAlpaca (Tang et al., 2023) as our train and test datasets. Without demonstrations, the base model solves only 42% of the problems. When provided with the appropriate demonstration $c$ for each prompt $x$, the teacher achieves a 100% success rate. To further test reward proximity, we manually inspected 50 teacher reasoning traces. In all cases, not only were the final tool calls correct, but the intermediate chain-of-thought was valid and semantically grounded. These observations provide evidence for the first requirement, that the demonstration-conditioned model behaves as an optimal policy.

To verify the second requirement, we measure the KL divergence to the base policy $D_{\mathrm{KL}}(\pi \| \pi_0)$ as a proxy for the distance to $\pi_k$. We compare this divergence for both the SFT model trained on demonstrations and the demonstration-conditioned teacher. As shown in Figure 2 (right panel), the SFT model deviates substantially from the base model (1.26 nats), whereas the teacher remains significantly closer (0.68 nats)—nearly half the divergence. This validates that the teacher produces high-quality outputs while maintaining proximity to the base policy.

## 4. Experiments

We evaluate our method in two settings that reflect common forms of post-training adaptation: *Skill Learning* and *Knowledge Acquisition*.

In *Skill Learning*, we study whether a pretrained LLM with broad capabilities can acquire a new skill without degrading its existing abilities. We choose to experiment with tasks the models had not already been explicitly fine-tuned on (unlike math or coding) to show the benefits of continual learning. Therefore, we test our method on three domains:

- *Science Q&A*: Undergraduate-level scientific reasoning,

using Chemistry L-3 subset of SciKnowEval (Feng et al., 2024).
- *Tool Use*: Mapping a tool-API specification and user request to a tool call, using ToolAlpaca (Tang et al., 2023).
- *Medical*: Clinical reasoning questions, with training data from stage 1 of the HuatuoGPT-o1 pipeline and evaluation from stage 2 (Chen et al., 2024).

In *Knowledge Acquisition*, the objective is different: the model must integrate genuinely new factual content not present in its pretraining data. We construct a corpus of Wikipedia articles describing natural disasters that occurred in 2025 (after the training knowledge cutoff), totaling approximately 200K tokens. Following Mecklenburg et al. (2024), we generate question–answer pairs about these articles (such as "Which regions were affected by the 2025 Myanmar earthquake?"), yielding an SFT dataset roughly 5× larger than the source corpus. This setting tests whether the model can absorb newly injected knowledge rather than merely improving skills it already has.

**Evaluation.** We evaluate two primary metrics:

- *In-Distribution Accuracy:* Accuracy on held-out test data for the newly introduced task. For Knowledge Acquisition, we use two variants: (1) All details correct (Strict Accuracy). (2) The answer contains correct information and no incorrect statements (Lenient Accuracy).
- *Previous Capabilities:* Performance on a suite of established benchmarks that probe general reasoning and world knowledge: HellaSwag (Zellers et al., 2019), TruthfulQA (Lin et al., 2021), MMLU (Hendrycks et al., 2020), IFEval (Zhou et al., 2023), Winogrande (Sakaguchi et al., 2021), and HumanEval (Chen et al., 2021). We report the average performance across these datasets as a measure of catastrophic forgetting.

For the *Knowledge Acquisition* setting, we include a third

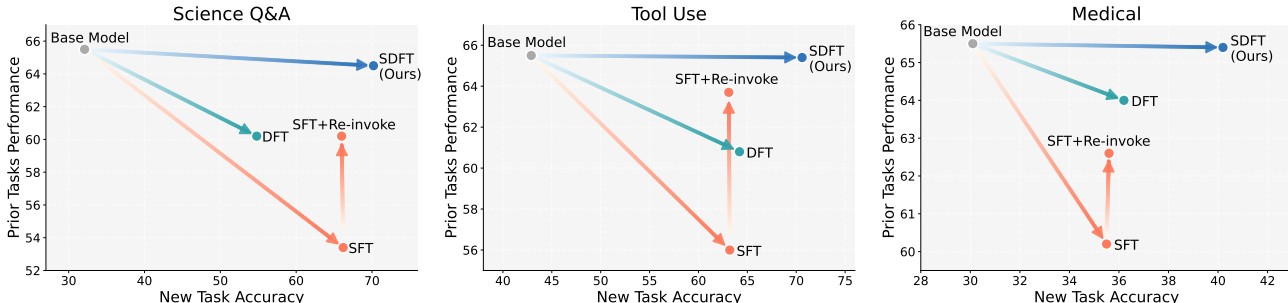

*Figure 4.* Performance trade-offs between new task accuracy and retention of prior capabilities. Each point represents a trained model, with the top-right indicating ideal performance (high accuracy on both new and previous tasks). SDFT consistently achieves superior Pareto efficiency compared to baselines across all three skill learning tasks.

metric:

- *Out-of-Distribution Accuracy:* "Indirect" questions whose answers depend on the injected knowledge but do not directly reference it (e.g., "Which countries required international humanitarian aid in 2025?"). This measures whether the new information has been properly integrated into the model's internal memory rather than memorized in a narrow form.

**Baselines.** In the *Skill Learning* setting, we compare our method to standard **SFT** and to **DFT** (Wu et al., 2025b), which uses importance sampling to treat the offline dataset as on-policy samples. We also include the recently proposed "**Re-invocation**" method (Lu & Lab, 2025), which performs additional on-policy distillation from the base policy on general-purpose prompts after SFT to restore prior capabilities.

In the *Knowledge Acquisition* setting, we compare our method to **CPT** (Continual Pre-Training), which trains directly on the text corpus using next token prediction loss and **SFT**, which trains on the question-answer pairs. In addition, we also compare with pure ICL methods. Because the full corpus exceeds the model's context window, we evaluate **RAG** with an oracle retriever that always provides the correct article for each question.

Unless otherwise noted, all experiments were performed on the Qwen2.5-7B-Instruct model. For each baseline, we perform a hyperparameter sweep and report results for the model achieving the highest validation performance on the target task. Full training details, see Appendix B.

### 4.1. On-policy learning leads to better generalization

Prior work has shown that on-policy learning achieves better in-distribution performance (Ross et al., 2011), as well as superior out-of-distribution generalization (Chu et al., 2025). We investigate whether these advantages also arise in our on-policy distillation framework. For that, we measure performance on test set on all our training tasks, as well as OOD generalization in the Knowledge Acquisition setting.

**Results.** Results for *Skill Learning*, as shown in Figure 4, indicate that our method achieves higher new-task accuracy than SFT, which represents better in-distribution generalization. We attribute these gains to the fact that off-policy learning trains only on expert-induced trajectories; errors at test-time can push the policy into unseen states, causing compounding errors. On-policy imitation learning avoids this mismatch by training on the state distribution induced by the learned policy itself (Ross et al., 2011).

The results for *Knowledge Acquisition* appear in Table 1. Since the new knowledge was not included in the base model's training, it cannot answer any of the questions correctly. Consistent with earlier observations (Mecklenburg et al., 2024), continual pretraining performs poorly. SFT on questions improves performance substantially but still lags behind our SDFT. On strict accuracy, it reaches 80% while our on-policy method achieves 89% and nearly closes the gap to the oracle RAG model. The advantage becomes even clearer on out-of-distribution questions, where our method achieves close to perfect accuracy, while SFT's performance remains low. This disparity underscores a key limitation of SFT: it teaches the model to reproduce specific answers but does not reliably incorporate the underlying facts into the model's broader knowledge base.

|  | Accuracy (strict) | Accuracy (lenient) | OOD Accuracy |
|---|---|---|---|
| Base | 0 | 0 | 0 |
| Oracle RAG | 91 | 100 | 100 |
| CPT | 9 | 37 | 7 |
| SFT | 80 | 95 | 80 |
| SDFT (Ours) | **89** | **100** | **98** |

*Table 1.* SDFT effectively integrates new factual knowledge, thus achieving better accuracy both in- and out-of-distribution.

Finally, with on-policy RL there is a concern for superficial improvements through entropy reduction rather than acquisition of new behaviors (Yue et al., 2025; Wu et al.,

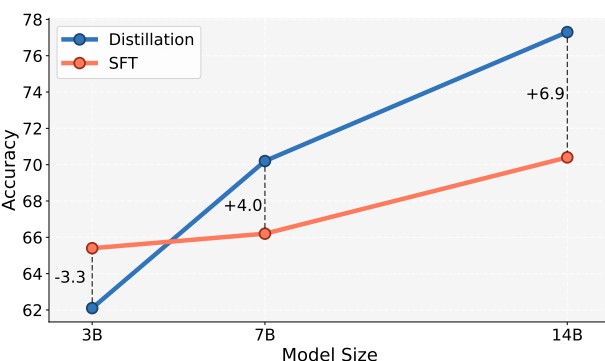 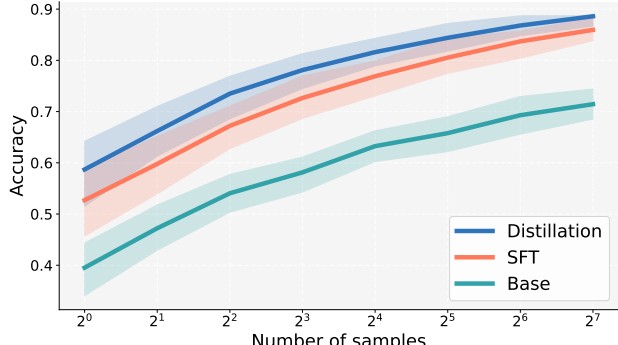

*Figure 5.* (**Left**) SDFT benefits from model scale. Performance gap between SDFT and SFT on the Science Q&A task increases with model size, as larger models have stronger in-context learning capabilities. (**Right**) SDFT improves pass@k across various k, indicating genuine skill acquisition rather than entropy collapse.

2025a). To ensure our gains are not merely due to distributional sharpening, we evaluate pass@$k$ for $k$ up to 128 in the Skill Learning Setting. As shown in Figure 5 (right), the performance gains over both the base model and SFT persist uniformly across all $k$. This indicates that the improvements reflect genuine skill acquisition rather than entropy collapse.

### 4.2. Learning without forgetting

A central claim of SDFT is that, due to its on-policy nature, it can acquire new skills while mitigating catastrophic forgetting. To test this, we perform the following experiments:

1. **Single Task Learning.** A convenient case study for continual learning is fine-tuning a model on a single task. Using the *Skill Learning* setting, we compare the broad capabilities of our models before and after training on each task.

2. **Multi-Task Continual Learning.** We investigate a more complex continual learning experiment in which a single model is trained sequentially on each task. The goal here is to measure catastrophic forgetting over longer training and to see whether the model retains the capabilities it learned at each stage of training.

**Results.** The results for single-task training, presented in Figure 4, show that our method is the only approach to improve performance on the new task without significant degradation in prior capabilities. In contrast, standard SFT produces substantial catastrophic forgetting across all evaluated benchmarks. Augmenting SFT with the "re-invoke" procedure partially restores lost abilities but does not recover the base model's full capabilities. DFT, which performs approximate on-policy updates, exhibits reduced forgetting relative to SFT but still results in noticeable degradation. For the breakdown of the score over prior tasks, see Table 5.

We now turn to the more challenging setting of long-horizon continual learning, where a single model is trained sequentially on all three skills. Figure 3 shows that SDFT enables stable accumulation of skills over time. As training progresses, the model improves on each newly introduced task while maintaining performance on previously learned ones. In contrast, SFT exhibits severe interference—performance on earlier skills rapidly degrades once training shifts to a new task, resulting in oscillatory behavior rather than cumulative learning. This demonstrates that SDFT supports true continual learning, allowing a single model to incrementally acquire multiple skills without catastrophic forgetting.

### 4.3. Effect of model size

Our method relies on the model's in-context learning (ICL) ability. As the teacher signal is generated by the model conditioned on demonstrations, and its quality depends on how well the model can interpret and extrapolate from them. Since ICL is known to improve with scale (Brown et al., 2020), we hypothesize that larger models yield stronger teacher policies and more effective SDFT updates. We test this via a scaling study on the Science Q&A task using multiple sizes from the Qwen-2.5 family (Hui et al., 2024).

**Results.** Figure 5 (left) shows a clear scaling trend. At small scales (e.g., 3B), weak ICL leads to poor teacher guidance and performance below SFT. As model size increases, SDFT gains grow monotonically–the 7B model improves by four points over SFT, and the 14B model by seven points. These results indicate that SDFT's effectiveness is tightly coupled to ICL strength and is likely to further improve at larger scales.

### 4.4. Training reasoning models without reasoning data

A major practical challenge in post-training reasoning models is the construction of high-quality supervision. Supervised fine-tuning for reasoning models typically requires access to intermediate reasoning traces, which are expensive to collect from human annotators and often unavailable from closed-source models that expose only final answers. As a result, many real-world datasets provide only the final

answer, without the full chain of thought.

Applying SFT to such data can be harmful for reasoning-capable models. When demonstrations include only short answers or abbreviated reasoning, SFT suppresses the model's existing long chain-of-thought behavior by directly optimizing for output matching. For instance, a model that naturally produces long reasoning traces may be penalized for doing so when SFTed on concise solutions, leading to a collapse in reasoning depth. We hypothesize that on-policy self-distillation avoids this failure mode: because the student matches a demonstration-conditioned teacher derived from the same model, the supervision preserves the model's reasoning style even when only final answers are available.

We test this hypothesis using Olmo-3-7B-Think (Olmo et al., 2025), fine-tuned on the medical task described earlier, which contains no explicit chain-of-thought annotations. We compare standard SFT and our method in terms of task accuracy and average generation length, used as a proxy for retained reasoning depth.

**Results.** As seen in Table 2, standard SFT substantially degrades performance, reducing accuracy by nine points and sharply shortening responses, indicating a collapse in reasoning behavior. In contrast, our method improves accuracy, reaching 43.7%. These results demonstrate that our approach enables effective task learning for reasoning models even in the absence of explicit reasoning data.

|  | Accuracy | Avg. # of tokens |
|---|---|---|
| Olmo-3-7B-Think | 31.2 | 4612 |
| + SFT | 23.5 | 3273 |
| + SDFT (Ours) | 43.7 | 4180 |

*Table 2.* Training reasoning models without reasoning supervision. SFT degrades both task performance and general reasoning behavior (indicated by shortened responses). SDFT avoids this collapse by learning from a demonstration-conditioned teacher rather than directly from the demonstrations.

### 4.5. What drives the improvement in performance?

Our method combines two ingredients: a demonstration-conditioned teacher policy and an on-policy distillation objective. In Subsection 3.2, we validated that the conditioned model is a high-quality teacher—producing correct outputs while remaining close to the base policy. A natural question then arises: *if such a teacher already exists, is on-policy learning necessary, or would standard distillation suffice?*

To isolate the source of the performance gains, we use Tool Use task and compare our full algorithm against two alternative ways to use the teacher: (1) SFT from the teacher, where the student is trained offline to imitate samples generated by the teacher. (2) Offline distillation from the teacher, where the student minimizes a KL loss on a fixed dataset of

teacher-generated outputs (Mitra & Ulukus, 2025).

**Results.** As shown in Figure 6, neither form of offline distillation matches the performance of our on-policy approach. While distillation from the teacher improves over standard SFT, it consistently underperforms our method. This gap indicates that the benefits of SDFT cannot be attributed solely to the quality of the teacher and further highlights the importance of on-policy learning.

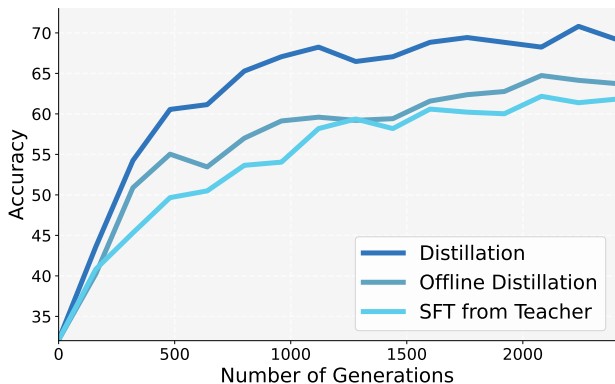

*Figure 6.* While offline distillation from the improves over standard SFT, it consistently underperforms on-policy SDFT, demonstrating that the benefits cannot be attributed solely to teacher quality.

## 5. Discussion and Limitations

**Relationship to on-policy RL.** SDFT is not an alternative to on-policy RL, but addresses a complementary learning regime. The two methods operate under different assumptions: SDFT applies when learning from expert demonstrations without access to an explicit reward function, whereas on-policy RL assumes a reward signal. Importantly, the two approaches can be naturally combined. As shown in Figure 5, SDFT consistently improves pass@$k$, indicating that it increases the diversity of high-quality generations, making it an effective initialization for subsequent RL training. If one nevertheless compares the training dynamics, a practical advantage of SDFT is its efficiency. It requires only a single on-policy generation per prompt, whereas many on-policy RL methods (e.g., GRPO) rely on group-based sampling to estimate relative advantages, substantially increasing generation cost. In addition, SDFT provides token- (or logit-) level supervision, yielding denser credit assignment than the trajectory-level advantages used in GRPO.

**Computational Costs.** A practical consideration when adopting SDFT is that, unlike standard SFT, it requires generating on-policy rollouts, resulting in approximately 2.5× FLOPs and roughly 4× the wall-clock training time compared to SFT. However, this additional cost must be viewed in context: many existing continual learning approaches, such as Re-invoke, require sequential training stages—first performing SFT, then conducting additional on-policy training to restore degraded capabilities. When accounting for

multi-stage process, SDFT can actually reduce total training time while simultaneously achieving better performance.

**Learned Artifacts.** A subtle failure mode of our approach is that the student can inherit spurious linguistic patterns from the teacher ("Based on the text..."). Empirically, we find that masking the loss over the first few tokens during training effectively suppresses these artifacts without harming downstream accuracy. While this workaround is effective in practice, it is fundamentally a heuristic fix. A more principled solution remains an open problem.

**Future Work.** Several promising directions remain for extending SDFT. First, although on-policy learning substantially reduces catastrophic forgetting compared to off-policy methods, some degradation of prior capabilities remains. Developing complementary techniques that further minimize forgetting represents an important avenue for future research. In addition, while we focus on expert demonstrations, extending SDFT to learn from non-expert or noisy demonstrations, or from unstructured data such as user conversations, would broaden its applicability to real-world settings where high-quality supervision is scarce.

## Impact Statement

By allowing models to update on-policy without requiring explicit reward functions, SDFT lowers a key barrier to safely adapting deployed foundation models as new tasks and information arise. This capability has positive implications for applications that require long-lived systems—such as scientific assistants, medical decision support, and robotics—where preserving prior knowledge while incorporating new expertise is critical. At the same time, enabling more effective post-deployment adaptation raises risks if such methods are applied to harmful tasks or used to rapidly repurpose models without appropriate oversight. While SDFT does not introduce fundamentally new misuse vectors beyond existing fine-tuning techniques, it may amplify the speed and persistence with which model behaviors can change. We therefore emphasize the importance of pairing continual learning methods with robust evaluation, monitoring, and governance practices, particularly in high-stakes domains.

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

# A. Additional Ablations

## A.1. Estimating the KL gradient

A central component of our objective is the gradient of the KL divergence between the current policy $\pi_\theta(y|x)$ and the teacher policy $\pi(y|x, c)$. For sequence models, the KL divergence is defined at the sequence level as:

$$\mathrm{KL}(\pi_\theta || \pi) = \mathbb{E}_{y \sim \pi_\theta} \left[ \log \frac{\pi_\theta(y|x)}{\pi(y|x, c)} \right]$$

where $y = (y_1, \ldots, y_T)$ is generated autoregressively. Differentiating this quantity is non-trivial because $\pi_\theta$ appears both in the sampling distribution and inside the logarithm, and different practical estimators trade off bias, variance, and computational cost (Tang & Munos, 2025).

We consider and ablate several commonly used KL gradient estimators.

**Token-level (partial) estimator** . A widely used approximation decomposes the KL into token-level terms and differentiates each independently:

$$\widehat{g}_{\text{token}} = \sum_{t=1}^{T} \log \frac{\pi_\theta(y_t|y_{<t}, x)}{\pi(y_t|y_{<t}, x, c)} \nabla_\theta \log \pi_\theta(y_t|y_{<t}, x)$$

As shown in recent analyses, this estimator corresponds to a *partial* derivative of the sequence-level KL: it ignores the effect of early tokens on future token distributions and is therefore biased with respect to the true gradient (Tang & Munos, 2025).

**Full analytic per-token estimator.** An alternative is to compute the KL analytically at each timestep by marginalizing over the vocabulary:

$$\widehat{g}_{\text{analytic}} = \sum_{t=1}^{T} \sum_{v \in \mathcal{V}} \log \frac{\pi_\theta(v|y_{<t}, x)}{\pi(v|y_{<t}, x, c)} \nabla_\theta \log \pi_\theta(v|y_{<t}, x)$$

This estimator has strictly lower variance than sample-based token estimators, but it remains biased at the *sequence* level, since it still does not account for how the choice of $y_t$ influences future states $y_{>t}$. Despite this bias, it is often computationally attractive because it leverages quantities already produced during the forward pass.

**Rao-Blackwellized estimator.** Following recent work (Amini et al., 2025), one can further reduce variance by Rao-Blackwellizing the KL estimator, analytically integrating over next-token distributions while retaining Monte-Carlo sampling over prefixes. This yields an unbiased estimator of both the KL and its gradient with provably lower variance than standard Monte-Carlo estimators . However, this estimator is more expensive to compute.

$$\widehat{g}_{\text{rb}} = \sum_{t=1}^{T} \left[ \sum_{v \in \mathcal{V}} \log \frac{\pi_\theta(v|y_{<t}, x)}{\pi(v|y_{<t}, x, c)} \nabla_\theta \log \pi_\theta(v|y_{<t}, x) + k_\theta(y_{<t}) \sum_{i=1}^{t-1} \nabla_\theta \log \pi_\theta(y_i|y_{<i}, x) \right]$$

Where $k_\theta(y_{<t})$ is the stepwise KL term $\mathrm{KL}(\pi_\theta(\cdot|y_{<t}, x) || \pi(\cdot|y_{<t}, x, x))$.

We empirically ablate all three estimators in our training pipeline. Despite its theoretical bias, we find that the *full analytic per-token estimator* consistently yields the most stable optimization and best downstream performance. In contrast, the token-level estimator exhibits higher variance and weaker KL control, while the Rao–Blackwellized estimator did not provide measurable gains in our setting relative to its additional complexity.

We also experimented with drawing multiple trajectories per prompt to reduce variance in the gradient estimator, which is theoretically beneficial for Monte-Carlo estimates. In practice, however, increasing the number of samples per prompt produced negligible improvements while substantially increasing compute. As a result, we adopt a single-trajectory-per-prompt setup combined with the analytic per-token KL estimator in all main experiments.

## A.2. The Importance of Demonstration-Conditioned Context

We analyze which components of the teacher context are essential for the effectiveness of our method in the *Knowledge Acquisition* setting. Recent self-distillation approaches for knowledge injection perform *offline* distillation using only the

raw corpus as context (Eyuboglu et al., 2025; Kujanpää et al., 2025). In contrast, our approach differs along two dimensions: (i) the teacher is conditioned not only on the source text but also on a worked answer, and (ii) distillation is performed on-policy.

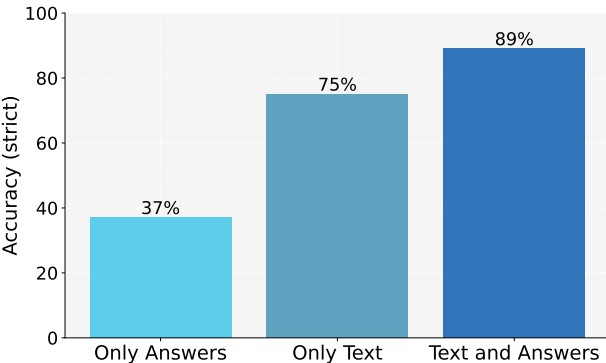

*Figure 7.* Conditioning the teacher on both article text and answer (89% strict accuracy) substantially outperforms text-only conditioning (75%), showing that the full demonstration context is critical for effective knowledge transfer.

In this subsection, we isolate the effect of the teacher context while holding the on-policy training procedure fixed. Specifically, we compare three variants: conditioning the teacher on *only the article text*, *only the answer*, and the full *text-plus-answer* context. A direct comparison to offline distillation methods is deferred to Section 4.5.

**Results.** The results are shown in Figure 7. Conditioning the teacher on the full text-plus-answer context yields the strongest performance, achieving 89% strict accuracy. Using only the article text substantially underperforms, consistent with prior findings that text-only distillation provides a weak and noisy supervisory signal. Conditioning on answers alone improves performance over text-only context but still falls short of the full context. These results suggest that answer-conditioned context plays a critical role by providing a stronger guidance for the student policy.

### A.3. Choice of teacher model

We ablate the choice of teacher policy used for distillation. While our framework does not require an external teacher, the stability of training depends critically on how the teacher is instantiated.

Using the frozen base model as the teacher yields stable training but consistently underperforms, as the teacher fails to reflect improvements acquired during learning. At the other extreme, using the student model itself as the teacher leads to severe instabilities. In this setting, small stochastic fluctuations in token-level probability updates can be rapidly amplified through the on-policy feedback loop, causing training to diverge. We find that maintaining an exponential moving average (EMA) of the student parameters provides an effective compromise. As shown in Figure 8, the EMA teacher tracks the student's progress while smoothing high-variance updates, resulting in both stable training and superior final performance.

## B. Training and Evaluation details

### B.1. Training Details

All experiments were conducted using the Hugging Face TRL library. Each experiment was conducted on a single NVIDIA H200 GPU. We perform full fine-tuning of the entire model's parameters. For each method, we performed a hyperparameter sweep over learning rates, batch sizes, and training epochs. We report test results for the model checkpoint that achieved the best validation performance on the target task.

Tables 4 and Y present the full hyperparameter search spaces and final selected values for the Skill Learning and Knowledge Acquisition settings, respectively. Across all tasks, we found that SDFT benefits from training for multiple epochs—typically 2 epochs for Skill Learning tasks and 4 epochs for Knowledge Acquisition. In contrast, SFT tends to overfit rapidly and showed no performance gains beyond a single epoch in most cases.

For SDFT, The teacher context was constructed using the prompt template shown in Section 3. We employed the analytic per-token KL gradient estimator (see Appendix A.1) with a single on-policy rollout per training example.

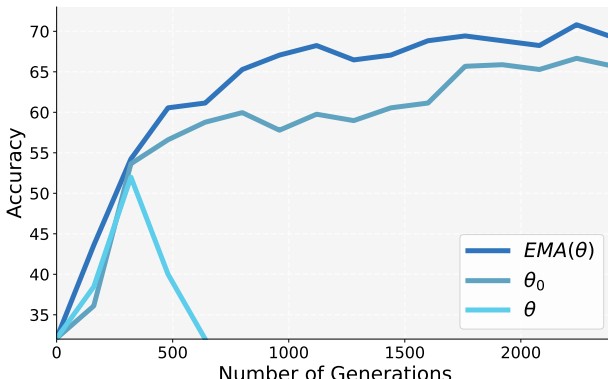

*Figure 8.* EMA teacher provides stable and effective training. Using the frozen base model, which fails to track learning progress lead to inferior results. Using the current student directly as the teacher leads to training instabilities.

| Hyperparameter | SFT | DFT | SDFT |
|---|---|---|---|
| Base Model | Qwen2.5 7B-Instruct | Qwen2.5 7B-Instruct | Qwen2.5 7B-Instruct |
| Learning Rate | {5e-6, 1e-5, 5e-5} | {5e-6, 1e-5, 5e-5} | {5e-6, 1e-5, 5e-5} |
| Optimizer | adamw | adamw | adamw |
| LR Scheduler | Cosine w. warmup | Cosine w. warmup | Cosine w. warmup |
| Warmup steps | 10 | 10 | 10 |
| Epochs | {1,2} | {1,2} | {1,2} |
| Batch Size | {16,32,64} | {16,32,64} | {16,32,64} |
| Max Grad Norm | 1 | 1 | 1 |
| bfloat16 | True | True | True |
| Weight Decay | 0 | 0 | 0 |
| *SDFT-only hyperparameters* | | | |
| EMA $\alpha$ | | | {0.01, 0.02, 0.05} |
| Max generation length | | | 2048 |

*Table 3.* Hyperparameters used for the Skill Learning experiments. Curly braces {} indicate a sweep over the specified values.

### B.2. Evaluation Details

**Sampling Strategy.** For accuracy metrics, we used greedy decoding (temperature = 0). For pass@k experiments, we used temperature = 1.0 with nucleus sampling (top-p = 0.95).

**Statistical Reporting.** Unless mentioned otherwise, all experiments were run over 3 random seeds. We report mean performance and 95% confidence intervals across seeds.

**Prior Capabilities Evaluation.** We assessed performance on general capabilities using the suite of benchmarks described in Section 4: HellaSwag (Zellers et al., 2019), TruthfulQA (Lin et al., 2021), MMLU (Hendrycks et al., 2020), IFEval (Zhou et al., 2023), Winogrande (Sakaguchi et al., 2021), and HumanEval (Chen et al., 2021). All benchmark evaluations were conducted using the Language Model Evaluation Harness (Gao et al., 2024).

### B.3. Dataset Details

**Science Q&A.** We used the Chemistry L-3 subset from SciKnowEval (Feng et al., 2024), splitting the data into approximately 75% train, 5% validation, and 20% test. To construct expert demonstrations, we queried GPT-4o, sampling up to 8 responses per prompt and retaining a single response that matched the correct final answer. This procedure yielded valid demonstrations for 100% of training examples. Since this is a multiple-choice dataset, accuracy was computed by exact match between the model's final answer choice and the ground truth.

| Hyperparameter | SFT | CPT | SDFT |
|---|---|---|---|
| Base Model | Qwen2.5 7B-Instruct | Qwen2.5 7B-Instruct | Qwen2.5 7B-Instruct |
| Learning Rate | {5e-6, 1e-5, 5e-5} | {1e-6, 5e-6, 1e-5} | {5e-6, 1e-5, 5e-5} |
| Optimizer | adamw | adamw | adamw |
| LR Scheduler | Cosine w. warmup | Cosine w. warmup | Cosine w. warmup |
| Warmup steps | 10 | 10 | 10 |
| Epochs | {1,2} | {1,2,4,8} | {1,2,4} |
| Batch Size | {16,32,64} | N/A | {16,32,64} |
| Max Grad Norm | 1 | 1 | 1 |
| bfloat16 | True | True | True |
| Weight Decay | 0 | 0 | 0 |
| *SDFT-only hyperparameters* | | | |
| EMA $\alpha$ | | | {0.01, 0.02, 0.05} |
| Max generation length | | | 1024 |

*Table 4.* Hyperparameters used for the Knowledge Acquisition experiments. Curly braces {} indicate a sweep over the specified values.

**Tool Use.** We used the ToolAlpaca dataset (Tang et al., 2023), following the original train-test split provided by the authors. Expert demonstrations were included in the original dataset. Accuracy was evaluated using regex matching against the ground-truth API call, accounting for variations in argument ordering.

**Medical.** We built upon the HuatuoGPT-o1 dataset (Chen et al., 2024), which provides both an SFT training set and a collection of problems with only final answer (used in the original paper for RL training). For training, we used only the English-language questions, yielding approximately 20,000 examples. For evaluation, we randomly sampled 1,000 verifiable questions from the verifiable problem set. Since these are open-ended clinical reasoning questions, we used GPT-5-mini as an automated evaluator with the following prompt:

```
You are an expert medical evaluator assessing whether a model's response
correctly answers a medical question. Your task is to compare the model's
response to the reference answer and determine if the model's response is:
1. CORRECT: The response contains the key medical information from the
reference answer, even if phrased differently or includes additional correct
medical details.
2. INCORRECT: The response is medically wrong, misses the main point, or
provides incorrect medical information.
Focus on medical accuracy and completeness, not on writing style or verbosity.

[Medical Question]
{question}
[Reference Answer]
{reference_answer}
[Model Response]
{model_response}
Evaluate the model's response. Output ONLY one of: "CORRECT" or "INCORRECT".
```

**Knowledge Acquisition.** We constructed a corpus of Wikipedia articles describing natural disasters that occurred in 2025 (after the model's knowledge cutoff), including:

2025 Myanmar earthquake, 2025 Kamchatka earthquake, 2025 Uttarakhand flash flood, Typhoon Kalmaegi, Tropical Storm Wipha, Cyclonic Ditwah, Hurricane Melissa, Kentwood Carson Tornado, July 2025 Central Texas floods.

Following Mecklenburg et al. (2024), we used GPT-5 to generate question-answer pairs from these articles, using the following prompt:

```
You are a helpful assistant that helps me write questions for an exam. You
will be given a wiki article and you will need to write 100 question on the
content of the wiki article. The question should require recalling multiple
pieces of information from the wiki article. Do not repeat the same question

The questions should be in the following format:
Question: <question>
Answer: <answern>
```

We also manually verify that the same question wasn't generated more than once. For evaluation, we used GPT-5-mini as an automated evaluator with the following prompt:

```
You are an expert evaluator assessing whether a model's response correctly
answers a question.

Your task is to compare the model's response to the reference answer and
determine if the model's response is:
1. CORRECT: The response contains the key information from the reference
answer, even if phrased differently or includes additional correct details.
2. PARTIALLY_CORRECT: The response contains most of the key information from
the reference answer but misses some details.
3. INCORRECT: The response is wrong, misses the main point, or provides
incorrect information.

Focus on factual accuracy and completeness, not on writing style or verbosity.

[Question]
{question}

[Reference Answer]
{reference_answer}

[Model Response]
{model_response}

Evaluate the model's response. Output ONLY one of:
"CORRECT", "PARTIALLY_CORRECT", or "INCORRECT".
```

---

**Algorithm 1** Self-Distillation Fine-Tuning (SDFT)

---

**Require:** Demonstration dataset $\mathcal{D} = \{(x_i, c_i)\}_{i=1}^{N}$
**Require:** Autoregressive model $\pi_\theta$; student context prompt $\text{Ctx}_S(x)$; teacher context prompt $\text{Ctx}_T(x, c)$
**Require:** Batch size $B$, max generation length $T$, learning rate $\eta$, teacher EMA rate $\alpha$
 1: Set teacher weights $\phi = \theta$.
 2: **for** each training step **do**
 3:     Sample minibatch $\mathcal{B} = \{(x_i, c_i)\}_{i=1}^{B} \sim \mathcal{D}$
 4:     **for all** $(x_i, c_i) \in \mathcal{B}$ **in parallel do**
 5:         **Student rollout (on-policy):**
 6:         $s_i \leftarrow \text{Ctx}_S(x_i)$
 7:         Sample $y_i = (y_{i,1:T}) \sim P_{\text{sample}}(\cdot \mid s_i)$
 8:         **Compute teacher and student token logprobs on the sampled tokens:**
 9:         $t_i \leftarrow \text{Ctx}_T(x_i, c_i)$
10:         Using TrainEngine, compute
11:             $\ell_{i,t}^{S} \leftarrow \log \pi_\theta(y_{i,t} \mid y_{i,<t}, s_i)$ and
12:             $\ell_{i,t}^{T} \leftarrow \log \pi_\phi(y_{i,t} \mid y_{i,<t}, t_i)$
13:     **end for**
14:     **Gradient computation and update:**
15:     Compute gradient estimate using Eq. A.1:
16:         $g \leftarrow \frac{1}{B} \sum_{i=1}^{B} g_{\text{analytic}}\Big(\{(\ell_{i,t}^{S}, \ell_{i,t}^{T})\}_{t=1}^{T}\Big)$
17:     If needed, add importance sampling to compensate for differences between the inference engine (e.g., VLLM) and the training code.
18:     Update parameters: $\theta \leftarrow \theta - \eta\, g$
19:     Update teacher parameters: $\phi \leftarrow \alpha\theta + (1 - \alpha)\phi$
20: **end for**

---

| | New Task: | | | Previous Tasks: | | | | |
|---|---|---|---|---|---|---|---|---|
| | Science Q&A | Hellaswag | Humaneval | IFeval | MMLU | TruthfulQA | Winogrande | Avg. |
| Base (Qwen2.5-7B) | 32.1 | **62.0** | 65.8 | **74.3** | **71.7** | **47.9** | 71.1 | **65.5** |
| SFT | 66.2 | 55.0 | 54.8 | 35.3 | 64.6 | 36.8 | **73.7** | 53.4 |
| SFT + re-invoke | 66.0 | 61.6 | 63.4 | 52.9 | 68.7 | 45.2 | 70.0 | 60.2 |
| DFT | 54.8 | 57.6 | 67.0 | 60.4 | 69.4 | 38.8 | 68.2 | 60.2 |
| SDFT (Ours) | **70.2** | 60.9 | **68.9** | 66.8 | 70.7 | 46.5 | 73.1 | 64.5 |

| | New Task: | | | Previous Tasks: | | | | |
|---|---|---|---|---|---|---|---|---|
| | Tooluse | Hellaswag | Humaneval | IFeval | MMLU | TruthfulQA | Winogrande | Avg. |
| Base (Qwen2.5-7B) | 42.9 | **62.0** | 65.8 | **74.3** | **71.7** | 47.9 | 71.1 | **65.5** |
| SFT | 63.2 | 57.3 | 50.0 | 49.8 | 70.2 | 37.5 | **73.1** | 56.0 |
| SFT + re-invoke | 63.1 | 61.7 | **68.9** | 59.1 | 71.5 | **49.1** | 71.6 | 63.7 |
| DFT | 64.2 | 59.7 | 61.4 | 60.2 | 71.6 | 40.2 | 71.5 | 60.8 |
| SDFT (Ours) | **70.6** | 61.6 | 68.3 | 71.9 | 71.5 | 47.3 | 71.7 | 65.4 |

| | New Task: | | | Previous Tasks: | | | | |
|---|---|---|---|---|---|---|---|---|
| | Medical | Hellaswag | Humaneval | IFeval | MMLU | TruthfulQA | Winogrande | Avg. |
| Base (Qwen2.5-7B) | 30.1 | **62.0** | 65.8 | 74.3 | **71.7** | **47.9** | 71.1 | **65.5** |
| SFT | 35.5 | 59.5 | 62.1 | 56.6 | 70.5 | 39.8 | **72.9** | 60.2 |
| SFT + re-invoke | 35.6 | 61.5 | 63.1 | 67.6 | 70.0 | 42.3 | 71.4 | 62.6 |
| DFT | 36.2 | 61.9 | 64.6 | **74.6** | 71.6 | 40.1 | 71.3 | 64.0 |
| SDFT (Ours) | **40.2** | 61.4 | 67.7 | 72.3 | 71.5 | 47.3 | 71.9 | 65.4 |

*Table 5.* The table reports the exact new-task accuracy and average prior-task performance for each method across all Skill Learning tasks.

