# OpenReview forum: "Self-Distillation Enables Continual Learning"
_ICML.cc/2026/Conference — ICML 2026 spotlight_

### Official Review · Reviewer_ETwN · 2026-02-21

**Soundness:** 2
**Presentation:** 3
**Significance:** 3
**Originality:** 2
**Overall Recommendation:** 4
**Confidence:** 3

**Summary:**

This paper proposes a continual learning algorithm for large language models (LLMs) that leverages a teacher model conditioned on expert demonstrations. By exploiting this conditioned teacher, the student performs on-policy learning, allowing it to acquire new tasks while mitigating catastrophic forgetting of previously learned knowledge. A key advantage of the approach is that the expert teacher is constructed solely from offline expert demonstrations, without requiring additional online interaction or expensive expert model training. The authors report empirical performance compared to several baselines across multiple experimental settings.

**Compliance With Llm Reviewing Policy:**

Affirmed.

**Final Justification:**

The paper presents a practically appealing approach to continual learning via ICL-conditioned teachers. The rebuttal clarified key concerns, particularly the 32B teacher comparison, which is encouraged to be included in the main paper. Overall, the rebuttal has addressed most of the raised issues.

**Key Questions For Authors:**

1. What are the key differences between the proposed method and Re-invocation-style approaches? Beyond using a conditioned expert teacher, how does the learning objective or training dynamic fundamentally differ?
2. How sensitive is the method to the EMA coefficient? Could you provide a sensitivity analysis or tuning details to demonstrate robustness and ensure fair comparison with baselines?
3. Since the paper positions the method as a continual learning algorithm, can you include comparisons with established continual learning approaches?
4. Why was the base model used as the teacher in Re-invocation instead of a stronger expert model, which is used in prior work? Would the conclusions change if a stronger teacher were used for that baseline?

**Limitations:**

yes

**Strengths And Weaknesses:**

**Strengths**

- The use of a conditioned teacher model is an interesting design choice. It allows the student to benefit from an on-policy learning regime while utilizing expert behavior derived purely from offline demonstrations.
- The paper provides extensive experimental results across multiple tasks and settings, suggesting consistent performance improvements over the considered baselines.
- Addressing continual learning in LLMs is an important and timely problem, and the proposed method is positioned toward practical training pipelines.

**Weaknesses**

- **Limited novelty relative to prior work**

    The proposed approach appears conceptually close to Re-invocation-style methods. Aside from replacing the teacher with a conditioned expert model, the overall training paradigm (on-policy distillation from a reference model to preserve prior capabilities) seems largely similar. The paper does not clearly articulate what fundamentally differentiates the proposed method from existing approaches at the algorithmic or theoretical level.

- **Unclear role of the ICL assumption**

    The method appears to rely on a strong assumption that conditioning on demonstrations produces a teacher policy that is effectively equivalent to a stable update of the base policy within a trust region. However, the discussion around the ICL assumption is not convincing. The stated validation (section 3.2) seem necessary but not sufficient for ensuring that the target optimal policy is approximated. Empirically, the gains may instead be attributable to using a stronger teacher (via conditioning), similar in effect to using a higher-quality teacher in Re-invocation.

- **Additional hyperparameter complexity (EMA coefficient)**

    The SDFT framework introduces additional EMA hyperparameter $alpha$, increasing the hyperparameter search space by three times compared to the baselines . The experimental section does not provide sensitivity analysis or ablation on these parameters, raising concerns about robustness and fairness.

- **Baseline selection and fairness concerns**

    Some baselines (e.g., Re-invocation, DFT) are not explicitly designed for continual learning. The paper claims advantages in a continual learning setting, yet comparisons with established continual learning methods (especially off-policy approaches) are missing. Additionally, the Re-invocation baseline appears to use the base model as the teacher, whereas prior work often employs a stronger external expert model. This choice may disadvantage the baseline and affect the fairness of comparisons.

---

> ### Author Rebuttal · Authors · 2026-03-28
>
> > The proposed approach appears conceptually close to Re-invocation-style methods. Aside from replacing the teacher with a conditioned expert model, the overall training paradigm (on-policy distillation from a reference model to preserve prior capabilities) seems largely similar.
>
> > What are the key differences between the proposed method and Re-invocation-style approaches? Beyond using a conditioned expert teacher, how does the learning objective or training dynamic fundamentally differ? Why was the base model used as the teacher in Re-invocation instead of a stronger expert model, which is used in prior work?
>
> SDFT and Re-Invoke address continual learning in fundamentally different ways:
> 1. In Re-Invoke, SFT is first performed on the new task, which leads to forgetting. To address this forgetting, Re-Invoke uses the initial model as a teacher to perform a second phase of training. This training is on-policy distillation with prompts that represent general model capabilities. Importantly, the goal of this second phase is not to improve performance compared to the base model but rather to restore it, and therefore, the initial model is the best fit and standard choice for this. If the reviewer can point us to works that use a different Re-Invoke approach, we would be happy to compare them.
>
> 2. SDFT, by contrast, is a single-stage method that prevents forgetting from occurring in the first place via on-policy learning, without any training on previous tasks.
>
> > The method appears to rely on a strong assumption that conditioning on demonstrations produces a teacher policy that is effectively equivalent to a stable update of the base policy within a trust region. However, the discussion around the ICL assumption is not convincing. The stated validation (section 3.2) seem necessary but not sufficient for ensuring that the target optimal policy is approximated. Empirically, the gains may instead be attributable to using a stronger teacher (via conditioning).
>
> We appreciate the reviewer raising this point, but we want to clarify that this point is actually the core of our contribution, not a limitation. The core idea of SDFT is that conditioning on demonstrations creates a high-quality teacher via ICL, and that this teacher, which is close to the base model's distribution, enables a stable on-policy update toward the target behavior.
> The reviewer's alternative hypothesis, that the gains come solely from teacher quality, independent of on-policy training, is directly tested in Section 4.5 (Figure 6). There, we compare SDFT against two forms of offline distillation using the exact same demonstration-conditioned teacher: SFT from teacher samples and offline KL distillation. Both consistently underperform SDFT despite having access to the same teacher signal. This demonstrates that a strong teacher is not enough, and that the on-policy training is necessary.
>
> > How sensitive is the method to the EMA coefficient?
>
> We ran a sweep over EMA coefficients on Science Q&A with Qwen2.5-7B-Instruct:
>
> | Coefficient | Performance |
> | --- | --- |
> | 0.0 | 66.3 |
> | 0.01 | 68.8 |
> | 0.05 | 70.4 |
> | 0.1 | 72.9 |
> | 0.2 | 72.6 |
> | 0.5 | 0.0 |
> | 1.0 | 0.0 |
>
> Performance is stable and consistently strong across a wide range of values (0.01–0.2), showing the method is not too sensitive. The collapse at high values (coeff≥0.5) is expected. At these values, the policy is kept too close to the student, making the EMA ineffective and giving rise to the same instabilities that led us to add EMA in the first place.
>
> > Since the paper positions the method as a continual learning algorithm, can you include comparisons with established continual learning approaches?
>
> Unfortunately, many well-established continual learning methods (e.g., EWC, LWF) are not applicable to LLM training. This is because in LLMs there isn’t a single definition of “previous task” which is needed for these methods. The model was trained through a series of training phases to perform any task that can be expressed in words.
>
> The most applicable continual learning baseline for our setting is data mixup (also known as Replay in the continual learning literature), where responses sampled from the base model on general-purpose prompts are mixed with the new-task SFT data during training. We ran this baseline on the Science Q&A task across four mixing ratios and report the results below:
>
> | Method | New Task Acc. | Forgetting |
> | --- | --- | --- |
> | SFT | +34.1 | -12.1 |
> | SDFT | +38.1 | -1.0 |
> | mixup 30% | +34.5 | -12.2 |
> | mixup 50% | +34.0 | -10.8 |
> | mixup 70% | +27.3 | -2.3 |
> | mixup 90% | +13.1 | -0.6 |
>
> Higher mixing ratios (70%, 90%) reduce forgetting relative to standard SFT, but at a substantial cost to new-task accuracy. Lower ratios preserve task performance but do not meaningfully reduce forgetting. In contrast, SDFT achieves higher new-task accuracy than any mixup ratio while also better preserving prior capabilities, demonstrating a more favorable tradeoff

---

> > ### Author Rebuttal · Reviewer_ETwN · 2026-04-01
> >
> > The rebuttal clarifies the structural difference from Re-Invoke (single-stage vs two-stage), but the core algorithmic mechanism remains the same: on-policy KL distillation from a teacher. Using ICL to condition the teacher is interesting, but it is essentially one way of constructing a stronger teacher. Without comparing against on-policy distillation with a stronger external expert as the teacher, which has proven effective in reasoning settings, it is difficult to tell whether the gains come from the ICL mechanism itself or simply from teacher quality. A direct comparison with a stronger-teacher variant is needed to substantiate the claimed contribution.
> >
> > ---
> > **(Reply to Authors' Reply Rebuttal Comment)**
> >
> > Thank the authors for the clarification and the additional 32B teacher comparison.
> > This result is informative, and it would strengthen the empirical justification in Section 3.2, which, as noted in the earlier review, currently provides necessary but not sufficient validation of the ICL assumption. Including this comparison in the main paper is encouraged.
> >
> > While the algorithmic novelty remains somewhat limited relative to existing on-policy distillation approaches, constructing a strong teacher purely through demonstration conditioning is practically appealing. The paper was enjoyable to read overall.
> > In light of the rebuttal, raising the score to 4.

---

> > > ### Author Response · Authors · 2026-04-02
> > >
> > > We thank the reviewer for their response, and are glad that the difference between our method and Re-Invoke is clearer now.
> > >
> > > As we’ve demonstrated in Section 3.2 the conditioned teacher is having 100% accuracy on the questions that are in the training dataset, making it an almost perfect teacher. For comparison, we performed on-policy distillation from the Qwen2.5-32B-Instruct, the largest model a single H200 GPU can support during training of the 7B model. Although much larger, the 32B model achieves 47.5% accuracy on the Science Q&A dataset, which is much better than the 33.3% accuracy of the 7B model but still far from perfect.
> > >
> > > After distillation, the 7B model matches the teacher at 47.3, lower than SDFT's performance. Distilling from a large, powerful teacher can be simpler than SDFT, but it requires access to one and enough GPUs to run it during the training. In contrast, the teacher created by conditioning on demonstrations is the same size as the student and can even outperform a larger non-conditioned model as a teacher.

---

### Official Review · Reviewer_Z7qo · 2026-03-10

**Soundness:** 3
**Presentation:** 4
**Significance:** 4
**Originality:** 3
**Overall Recommendation:** 5
**Confidence:** 4

**Summary:**

This paper introduces Self-Distillation Fine-Tuning (SDFT) as a method to fine tune LLMs with expert instructions. The idea is to have the same base LLM function as both a student and a teacher: Given only the current task as prompt, the student starts solving it. The teacher receives in its prompt additionally an expert solution to the task. The SDFT objective is to minimize the reverse KL-divergence between the student's and the teacher's next token distribution in the student's generated trajectory. In contrast to supervised fine-tuning directly on the expert demonstrations, this method is "on policy", since the teacher can provide token-wise corrections for the student's self-generated responses. This leads to improved continual learning, better performance on the new tasks and higher retention of past skills.

**Compliance With Llm Reviewing Policy:**

Affirmed.

**Final Justification:**

I thank the authors for their response, and maintain my positive assessment of the work.

**Key Questions For Authors:**

- The method relies on the in-context learning ability of the base model. In section 4.3, and experiment shows that hence—as we perhaps would expect—for a model size of 3B, SFT performs better than self-distillation. Is there any property beyond size that plays a role for how promising self-distillation is? Is there a way to train a base model in such a way to make it a better teacher?
- In Figure 3, it seems that SDFT converges faster than SFT (this is clear for "Train on Tooluse", for the other tasks it may be conflated by lower starting performance). Is this true in general? In the same figure, why is the number of gradient steps per task different for SDFT and SFT?
- Some figures include confidence intervals, but most don't. Is there a reason for that?

**Limitations:**

Yes

**Strengths And Weaknesses:**

**Strengths:**

I enjoyed this paper. The introduced method is intuitive and well explained. The experiments are sensibly chosen and very convincing. The whole paper seems polished. I think self distillation has the potential to be an impactful concept.

**Weaknesses**:
- The main experiments are done with only one model family (Qwen). I think it is reasonable to assume that the results will hold for other models, a confirmation of that would be nice.
- I understand the authors' decision to have another submission with a related topic, since here there is enough content and novelty for a conference paper. Still, I wonder if combining the two would not have led to an even stronger submission, because the core idea seems to be very much the same.

---

> ### Author Rebuttal · Authors · 2026-03-28
>
> > The main experiments are done with only one model family (Qwen). I think it is reasonable to assume that the results will hold for other models, a confirmation of that would be nice.
>
> We thank the reviewers for the question. We first want to point their attention to the experiment in Section 4.3, which was done on Olmo-3-7B-Think. In addition, to reassure that the results are not model-specific, we rerun the Science Q&A experiment from Figure 4 using Olmo-3-7B-Instruct (following the exact experimental protocol as the Qwen models):
>
> |  | New task score | Prior task score (Avg.) |
> | --- | --- | --- |
> | Base | 33.9 | 0.463 |
> | SFT | 65.9 | 0.426 |
> | SDFT | 66.7 | 0.461 |
>
> >  Is there any property beyond size that plays a role for how promising self-distillation is? Is there a way to train a base model in such a way to make it a better teacher?
>
> Besides size, newer models, even of the same size, have better ICL capabilities. For example, Qwen3 is much stronger compared to the original Qwen model. In addition, there are some works that suggested training methods to improve the ICL capabilities of models [1,2,3]
>
> > In Figure 3, it seems that SDFT converges faster than SFT (this is clear for "Train on Tooluse", for the other tasks it may be conflated by lower starting performance). Is this true in general? In the same figure, why is the number of gradient steps per task different for SDFT and SFT?
>
> In general, yes, SDFT enjoys denser supervision (full logits objective) compared to SFT and, as such, tends to converge faster. As shown in Figure 6, even SFT from the teacher converged more slowly than SDFT. Regarding the gradient steps, we found that each method has a different optimal batch size and since the dataset size is fixed, it results in a different number of gradient steps. For more details on hyperparameter tuning, see Appendix B.
>
> > Some figures include confidence intervals, but most don't. Is there a reason for that?
>
> We did not manage to run multiple seeds for all experiments before the initial submission deadline. We will make sure to add them all in the camera-ready version.
>
> [1] Min, Sewon, et al. "Metaicl: Learning to learn in context." *Proceedings of the 2022 conference of the North American chapter of the Association for Computational Linguistics: Human Language Technologies*. 2022.
>
> [2] Kumar, Aviral, et al. "Training language models to self-correct via reinforcement learning." *arXiv preprint arXiv:2409.12917* (2024).
>
> [3] Wang, Yihan, et al. "Preserving in-context learning ability in large language model fine-tuning." (2022).

---

> > ### Author Rebuttal · Reviewer_Z7qo · 2026-04-02
> >
> > I had only minor concerns to begin with, and they have been addressed. While I'm not an expert in continual learning, I believe the paper makes a valuable and practical contribution. In response to the point of comparing to the use of a separate stronger teacher model that other reviewers raise: While this is an interesting baseline, it requires the existence of and access to a stronger model, which this methods elegantly avoids.

---

### Official Review · Reviewer_m28Z · 2026-03-11

**Soundness:** 3
**Presentation:** 3
**Significance:** 3
**Originality:** 2
**Overall Recommendation:** 4
**Confidence:** 3

**Summary:**

This paper introduces Self-Distillation Fine-Tuning (SDFT), which uses the same model as both student and teacher, to achieve continual learning. In comparison with on-policy RL, SDFT does not require explicit reward signals; compared with off-policy SFT, SDFT learns from on-policy signals and reduce catastrophic forgetting. The paper shows that SDFT is also equivalent to Inverse RL under specific conditions. Experiments on several domains verify that SDFT consistantly outperforms baseline methods, reduces forgetting and improves generalization.

**Compliance With Llm Reviewing Policy:**

Affirmed.

**Final Justification:**

I maintain my original positive rating after reading the rebuttal and other reviews.

**Key Questions For Authors:**

- How does the proposed method compare to [1]? I feel this method should be compared with in experiments. [1] proposes to do on-policy distillation with the student's own generation, while the teacher model is a stronger model. I wonder whether it can achieve better performance when a teacher model is avaliable.
- How does the design of in-context learning prompt and demonstration quality/count influence the performance of SDFT?


[1] Agarwal, et al. On-Policy Distillation of Language Models: Learning from Self-Generated Mistakes.

**Limitations:**

Yes

**Strengths And Weaknesses:**

Strengths:
- The paper is well-written and easy to follow. The proposed method is simple yet effective.
- The idea of using demonstration and in-context learning as a teacher model, although not new, makes sense in the setting of continual learning.
- The connection between SDFT and IRL is insightful.

Weaknesses:
- The experiment is relatively simple, although covering several interesting topics. It would be nice to see experiment results with broader setups. For example, the main experiments are conducted with Qwen 2.5 models, while it would be nice to verify the observations on other families of models.
- The experiments mainly compare SDFT with off-policy methods (SFT and its variants), while it is not very clear how the method compared with RL in terms of generalization and performance. Although RL is based on explict rewards, it would still be interesting to perform the comparison to understand the pros and cons better.

---

> ### Author Rebuttal · Authors · 2026-03-28
>
> We are glad the reviewer found our proposed method simple yet effective
>
> > It would be nice to see experiment results with broader setups. For example, the main experiments are conducted with Qwen 2.5 models, while it would be nice to verify the observations on other families of models.
>
> We thank the reviewers for the question. We first want to point their attention to the experiment in Section 4.3, which was done on Olmo-3-7B-Think. In addition, to reassure that the results are not model-specific, we rerun the Science Q&A experiment from Figure 4 using Olmo-3-7B-Instruct (following the exact experimental protocol as the Qwen models):
>
> |  | New task score | Prior task score (Avg.) |
> | --- | --- | --- |
> | Base | 33.9 | 46.3 |
> | SFT | 65.9 | 42.6 |
> | SDFT | 66.7 | 46.1 |
>
> SDFT again improves new-task accuracy while preserving prior capabilities, consistent with the Qwen results
>
> > The experiments mainly compare SDFT with off-policy methods (SFT and its variants), while it is not very clear how the method compared with RL in terms of generalization and performance. Although RL is based on explict rewards, it would still be interesting to perform the comparison to understand the pros and cons better.
>
> First, we want to emphasize that SDFT and RL operate under different assumptions. SDFT applies when only demonstrations are available, while RL requires a reward function. That said, we agree that a direct empirical comparison is informative, and we performed GRPO training of Qwen2.5-7B-Instruct on the Science Q&A dataset.
>
> A key practical distinction is computational cost. GRPO requires multiple generations per prompt for advantage estimation, compared to SDFT's single generation per prompt, which, in our experiment, means 16x more compute and wall-clock training time. With this in mind, the comparison should be interpreted as a cost-performance tradeoff rather than a direct apples-to-apples benchmark.
>
> Due to time constraints, we were able to run only 3 hyperparameter configurations for GRPO, with the best achieving a test accuracy of 74.8%, compared to SDFT's 70.2%. This suggests that when a verifiable reward is available and compute is not a constraint, RL can achieve higher accuracy. However, SDFT closes much of this gap at a fraction of the cost, and, as shown in Figure 5, consistently improves pass@k, making it a strong initialization for subsequent RL training when both approaches are available.
>
> > How does the proposed method compare to [1]? I feel this method should be compared with in experiments. [1] proposes to do on-policy distillation with the student's own generation, while the teacher model is a stronger model. I wonder whether it can achieve better performance when a teacher model is avaliable.
>
> As we’ve demonstrated in Section 3.2 the conditioned teacher is having 100% accuracy on the questions that are in the training dataset, making it an almost perfect teacher. For comparison, we performed on-policy distillation from the Qwen2.5-32B-Instruct, the largest model a single H200 GPU can support during training of the 7B model. Although much larger, the 32B model achieves 47.5% accuracy on the Science Q&A dataset, which is much better than the 33.3% accuracy of the 7B model but still far from perfect.
>
> After distillation, the 7B model matches the teacher at 47.3, lower than SDFT's performance. Distilling from a large, powerful teacher can be simpler than SDFT, but it requires access to one and enough GPUs to run it during the training. In contrast, the teacher created by conditioning on demonstrations is the same size as the student and can even outperform a larger non-conditioned model as a teacher.
>
> > How does the design of in-context learning prompt and demonstration quality/count influence the performance of SDFT?
>
> Regarding the effect of the number of demonstrations on performance, since we use a single demonstration per training example, the number of training examples is equivalent to the number of demonstrations seen during training. We direct the reviewer to Figure 6, where the x-axis (number of generations) directly reflects this quantity, showing how performance scales with the amount of demonstration data.
>
> As for the prompt, we intentionally used the simplest prompt we could construct, as our goal was to evaluate the method itself rather than prompt engineering. We view prompt optimization as a promising direction for future work.
> $$$$
>
>
> If the reviewer has any further questions or requires additional experimental evidence, we will be happy to answer them.

---

> > ### Author Rebuttal · Reviewer_m28Z · 2026-04-05
> >
> > Thank you for the additional results and they make sense to me. It would be nice if the additional experiments and analysis can be incorporated into the updated paper.

---

> > > ### Author Response · Authors · 2026-04-05
> > >
> > > Thank you again for the thoughtful follow-up. Since our rebuttal and new experiments seem to have addressed your main concerns, we would be very grateful if you would consider updating your score to better reflect your current assessment. We cannot update the paper during rebuttal, but we will incorporate all of them into the final camera-ready version.

---

### Official Review · Reviewer_Vxnp · 2026-03-12

**Soundness:** 4
**Presentation:** 4
**Significance:** 4
**Originality:** 4
**Overall Recommendation:** 6
**Confidence:** 4

**Summary:**

This paper introduces Self-Distillation Fine-Tuning (SDFT), a method for continual learning from expert demonstrations that enables on-policy updates without requiring a reward function or a reward model. SDFT successfully exploits in-context learning capabilities of LLMs to parametrize student and teacher models from the same base model. The teacher receives the expert demonstration as part of its input, while the student does not. The SDFT objective consists in minimizing the KL divergence between the student and teacher policies when evaluated on trajectories generated by the student. Experiments on skill learning and knowledge acquisition tasks show improved new-task performance and reduced catastrophic forgetting compared to supervised fine-tuning baselines, as long as the base model is of sufficient scale.

**Compliance With Llm Reviewing Policy:**

Affirmed.

**Final Justification:**

The rebuttal satisfactorily addressed my minor concerns and reinforced my original positive assessment. I encourage the authors to incorporate to the manuscript the theoretical clarification provided in their rebuttal. I am maintaining my original maximal score.

**Key Questions For Authors:**

1. Could the authors detail the steps to go from equation on l199 lhs -> l211 lhs? I'm not sure I see how $\nabla_\theta \log \pi_k$ is being incorporated in the t-summation.

**Limitations:**

yes

**Strengths And Weaknesses:**

**Strengths**

**Clear motivation and well-defined problem setting.** The paper addresses continual learning in foundation models trained from demonstrations, an important problem given the difficulty of specifying reward functions in many real-world domains. An important concept discussed by this manuscript is the distinction between on-policy and off-policy learning when adapting models over time. The authors argue that off-policy supervised fine-tuning contributes to catastrophic forgetting, motivating the search for methods that preserve prior capabilities while incorporating new skills. This framing is clear and well motivated.

**Simple and intuitive methodological contribution.** The proposed Self-Distillation Fine-Tuning (SDFT) method is conceptually straightforward: a demonstration-conditioned version of the model acts as a teacher, while the student generates trajectories and learns via reverse-KL distillation. This design successfully leverages in-context learning abilities of modern LLMs to self-distill knowledge from expert demonstrations without forgetting existing capabilities. The formulation is clean and relatively easy to implement.

**Insightful theoretical justification.** The paper provides a useful theoretical interpretation of the proposed objective by establishing connections between SDFT, inverse RL and trust region policy optimisation.

**Strong empirical evaluation across settings.** The experimental section evaluates the method across both skill learning and knowledge acquisition tasks, including tool use, scientific reasoning, and medical reasoning. Results show consistent improvements in new-task accuracy while maintaining higher performance on prior capabilities compared to SFT baselines. The authors also evaluate multi-task sequential learning, demonstrating that SDFT allows a single model to accumulate multiple skills without severe performance regression. Additional experiments analyze how SDFT can be employed to train reasoning models without chain-of-thought demonstrations, which strengthens the applicability of the method.

**Insightful analysis and ablations.** The paper includes useful analyses validating the key assumptions behind the method. Of particular interest was the importance of using a sufficiently large base model to ensure the teacher is a good approximation of the optimal policy. The paper also includes ablations of different components of SDFT, such as not employing on-policy trajectories for distillation and replacing the distillation objective by imitation learning on the teacher's generated trajectories.


**Weaknesses**

**(Mixed strenght/weakness) Reliance on strong base model.** The effectiveness of SDFT depends on the in-context learning capabilities of the demonstration-conditioned teacher. The authors’ own scaling experiments indicate that performance improves substantially with model size, and smaller models underperform SFT. This may limit applicability in lower-capacity models or non-LLM settings. However, this also means that this publication provides valuable signal for the community, as similar ideas may have been tried in the past on smaller models and discarded too promptly.

**Computational cost compared to SFT.** SDFT requires on-policy generation during training, increasing computational cost relative to SFT. The authors acknowledge this but also make an unsubstantiated claim that Re-invoke could have even higher costs due to its multi-stage process. Can the authors include a quantitative analysis or approximation of the additional costs of Re-invoke/DFT/SDFT compared to SFT? In the case of Re-invoke, does spending additional compute result in stronger recovery on the prior task? This is of particular interest since it appears that Re-invoke converts the additional compute into recovering original capabilities without experiencing any degradation in new task capabilities achieved via SFT, see Figure 4.

Minor clarifications / typos:
- l289 lhs: RAG is not introduced in the text.
- l422 rhs: typo in opening quotes

---

> ### Author Rebuttal · Authors · 2026-03-28
>
> We thank Reviewer Vxnp for their review, and we are glad they found the paper insightful and that the results are strong.
>
> > **(Mixed strenght/weakness) Reliance on strong base model.** The effectiveness of SDFT depends on the in-context learning capabilities of the demonstration-conditioned teacher… This may limit applicability in lower-capacity models or non-LLM settings.
>
> We agree with the reviewer on this point, and even acknowledged it ourselves in the paper. We just want to add that newer models, even of the same size, have better ICL capabilities. For example, Qwen3 is much stronger compared to the original Qwen model. If this trend continues, SDFT will become relevant to more smaller models.
>
> > **Computational cost compared to SFT.** Can the authors include a quantitative analysis or approximation of the additional costs of Re-invoke/DFT/SDFT compared to SFT? In the case of Re-invoke, does spending additional compute result in a stronger recovery on the prior task?
>
> We thank the reviewer for their question, as this is an important comparison. We included a cost comparison between SDFT and SFT in Section 5 and will now extend it to the other baselines. The following comparison is performed on the Science Q&A dataset, and the reported runtime is normalized relative to SFT. In addition, we vary the amount of Re-Invoke training we do to give a complete picture.
> Here's the table with both metrics:
>
> | Method | Wall Clock Runtime | Forgetting |
> |---|---|---|
> | SFT | 1x | -12.1 |
> | DFT | 1x | -5.2 |
> | SDFT | 4x | -1.0 |
> | SFT + Re-Invoke | 1x + 2x | -8.7 |
> | SFT + Re-Invoke (original exp.) | 1x + 4x | -5.3 |
> | SFT + Re-Invoke | 1x + 6x | -2.2 |
> | SFT + Re-Invoke | 1x + 8x | -0.8 |
> | SFT + Re-Invoke | 1x + 10x | -0.9 |
>
> As one can see, although it helps recover prior capabilities, Re-Invoke requires significantly more compute to achieve the same level of forgetting. We attribute this to the two-stage training, which requires the model to relearn what it has forgotten, unlike SDFT, which learns directly without forgetting.
>
> > Could the authors detail the steps to go from equation on l199 lhs -> l211 lhs?
>
> The way we wrote it, it does not go directly into the sum. An exact derivation results in each token-wise gradient being multiplied by a discounted sum of future reward:
>
> $$
> \mathbb E_{y\sim\pi_k}\left[\sum_{t=1}^T\nabla_\theta\log\pi_k(y_t|y_{<t},x)\sum_{t'=t}^T\gamma^{t'-t}r_t(y_t'|y_{<t'},x)\right]
> $$
>
> A choice of $\gamma=0$ will result in l211 and a choice of $\gamma=1$ in the Rao-Blackwellized estimator (see Appendix A.1). Both are valid estimators of the KL objective.
>
> We tried to simplify the derivation, but ended up making it unclear to the reader. We thank the reviewer for pointing it out, and we will revise it in the camera-ready version.

---

> > ### Author Rebuttal · Reviewer_Vxnp · 2026-04-03
> >
> > I only had a few minor concerns initially, and those have now been resolved. The additional analysis addresses the weakness pointed out on computational costs in my review: while Re-Invoke can match SDFT's performance, it requires expanding over twice as much compute. For completeness the "SFT + Re-Invoke 1x + 8x" result should be included in the revised manuscript. Regarding correctness of theoretical results, including the authors' provided clarification to the revised manuscript should be sufficient to explain the derivation step from l199 lhs to l211 lhs.
> >
> > I've already gave the maximal score for this paper in my original review and will keep it.

---

### Decision · Program_Chairs · 2026-04-30

**Decision:**

Accept (spotlight)

**Comment:**

The paper contributes a novel framework for continual learning that leverages a model's own in-context learning capabilities to act as its own teacher. There was a slight spread in initial scores ranging from weak accept to strong accept, reviewers uninimously agreed that the method offers a highly practical and effective approach to continual learning without catastrophic forgetting. Reviewers appreciated the robust empirical results across multiple domains like tool use and medical reasoning. Some concerns were initially raised regarding computational cost, baseline selection, and dependence on base model scale. However, the authors provided a strong rebuttal that addressed these concerns and demonstrated clear cost tradeoffs against RL baselines and ablation results comparing the conditioned teacher against simply using a larger 32B external model. I recommend acceptance.